

# Minimizing confounders and increasing data quality in murine models for studies of the gut microbiome

Jun Miyoshi[1], Vanessa Leone[1], Kentaro Nobutani[1], Mark W. Musch[1], Kristina Martinez-Guryn[1,2], Yunwei Wang[1], Sawako Miyoshi[1], Alexandria M. Bobe[1], A. Murat Eren[1] and Eugene B. Chang[1]

[1] Department of Medicine, The University of Chicago, Chicago, IL, United States of America
[2] Biomedical Sciences Program, Midwestern University, Downers Grove, IL, United States of America

## ABSTRACT

Murine models are widely used to explore host-microbe interactions because of the challenges and limitations inherent to human studies. However, microbiome studies in murine models are not without their nuances. Inter-individual variations in gut microbiota are frequent even in animals housed within the same room. We therefore sought to find an efficient and effective standard operating procedure (SOP) to minimize these effects to improve consistency and reproducibility in murine microbiota studies. Mice were housed in a single room under specific-pathogen free conditions. Soiled cage bedding was routinely mixed weekly and distributed among all cages from weaning (three weeks old) until the onset of the study. Females and males were separated by sex and group-housed (up to five mice/cage) at weaning. 16S rRNA gene analyses of fecal samples showed that this protocol significantly reduced pre-study variability of gut microbiota amongst animals compared to other conventional measures used to normalize microbiota when large experimental cohorts have been required. A significant and consistent effect size was observed in gut microbiota when mice were switched from regular chow to purified diet in both sexes. However, sex and aging appeared to be independent drivers of gut microbial assemblage and should be taken into account in studies of this nature. In summary, we report a practical and effective pre-study SOP for normalizing the gut microbiome of murine cohorts that minimizes inter-individual variability and resolves co-housing problems inherent to male mice. This SOP may increase quality, rigor, and reproducibility of data acquisition and analysis.

Corresponding author
Eugene B. Chang,
echang@medicine.bsd.uchicago.edu

## INTRODUCTION

The involvement of the gut microbiota in human diseases is under extensive investigation and has been boosted by recent advances in cultivation-independent bioinformatic approaches to study complex microbial communities. Reduced cost and increased access to 16S rRNA gene amplicon sequencing and metagenomic analysis to examine the bacterial community composition and function of the gut microbiota have improved our
understanding of the role of gut microbiota in health and disease. Epidemiological studies using these methodologies have presented the association of gut dysbiosis with various diseases ranging from intestinal diseases to extra-intestinal and systemic diseases (*Carding et al., 2015*; *Shreiner, Kao & Young, 2015*). Despite these advances, a large variation of microbiota has been observed between human subjects (*Faith et al., 2013*; *Lozupone et al., 2012*; *Turnbaugh et al., 2009a*; *Yatsunenko et al., 2012*). Human microbiota can be affected by factors, such as environment (*Yatsunenko et al., 2012*), diet (*David et al., 2014*; *Kashtanova et al., 2016*; *Wu et al., 2011*), sex (*Haro et al., 2016*) and age (*O'Toole & Jeffery, 2015*; *Saraswati & Sitaraman, 2014*). The resulting large interindividual variability in microbiota coupled with genetic diversity are confounding factors that are challenging to overcome when studying host-microbe interactions in humans. In contrast, many of these variables can be controlled when murine models are used in the study of the gut microbiota. While inherent differences between human and mouse are indeed limiting factors, a properly controlled murine study can provide important insights into host-microbe relationships which can help establish causality, disease pathogenesis, and interventional strategies. As in humans, the murine microbiota is influenced by various factors. In mice, these include breeding environment (facilities and rooms) (*Ericsson et al., 2015*; *Hufeldt et al., 2010*; *Rogers et al., 2014*), genetic backgrounds (strains) (*Org et al., 2016*; *Org et al., 2015*), diet (*Liu et al., 2012*; *Turnbaugh et al., 2009b*), sex (*Markle et al., 2013*; *Org et al., 2016*) and age (*Langille et al., 2014*; *Schloss et al., 2012*). Furthermore, even in one specific room within a specific pathogen-free (SPF) animal vivarium, variation between cages can be observed, a phenomenon called "cage effect" (*Hildebrand et al., 2013*; *McCafferty et al., 2013*; *Hoy et al., 2015*). In studies with large cohorts of mice that consist of multiple cages per group, cage effects can lead to a large variability in gut microbiomes among animals within a study. This variability at experimental onset immediately introduces artifacts into studies of the gut microbiota, which can mask and skew experimental findings related to metabolism (*Murphy et al., 2010*), the immune system (*Ivanov et al., 2008*; *Moon et al., 2015*) or disease activity (*Devkota et al., 2012*; *Markle et al., 2013*).

Several strategies have been used to decrease inter-individual variability of the gut microbiota. Co-housing approaches are used to enable transfer of microbiota through shared environment and coprophagia, however, this is limited by the number of animals that can be housed in a single cage (*McCafferty et al., 2013*). Often, to achieve the numbers of mice for meaningful statistical analyses and reproducibility, age-matched rather than littermate mice are used which are divided among treatments or cages. This practice reintroduces cage effects and other variables that impact starting microbiota. Oral gavage of microbiota has also been used to conventionalize and normalize germ-free and SPF mice with target microbial communities, however, this labor-intensive procedure induces stress and can cause injury or fatality even when performed by experienced personnel (*Arantes-Rodrigues et al., 2012*). For murine microbiota studies requiring multiple groups with a large number of animals, two simple measures have been used to prepare age-matched animals, including: (1) setting up multiple breeding pairs within a specific vivarium room where environment, cage changes, and dietary schedules are more uniform. Pups from different breeding pairs are then combined and used as a single group, and (2) a one-time

purchase of mice from a vendor, followed by acclimation in a specific vivarium room several weeks in advance of an intended study. Additional limitations of these current practices, including aggressive behavior in adult males when cohoused, variation in the gut microbiota of different breeding pairs, known generational drift in microbiota (*Choo et al., 2017*), and differences between batches of mice purchased from a vendor, can have profound effects on microbial membership and function that affect experimental outcomes.

Despite these attempts to normalize gut microbes, we have observed appreciable differences in starting gut microbiota. Indeed, in our own hands, we have encountered difficulty in reproducing clear microbiome data in murine studies due, in part, to large variability among mice at the study onset. Therefore, an unmet need is the development of a simple and effective standard operational procedure (SOP) to minimize the individual variability and cage differences of microbiota among mice at the onset of a study. The SOP should provide optimal conditions so that true effects of a specific treatment on the gut microbiota can be observed. At the same time, the approach should be applicable to studies where large numbers of mice and cages are required to sufficiently power a study and ensure reproducibility (*Moore & Stanley, 2016*). Considering these facts, we developed and vetted a "bedding transfer" procedure, where soiled bedding is mixed and distributed equally among pups at weaning (three weeks of age) until the start of a particular study (e.g., 6–12 weeks of age) among multiple cages of mice. This approach was contrasted with two conventional procedures used to minimize variability as described above, namely, in-house breeding without bedding transfer or mass animal purchase from a vendor followed by acclimatization. We hypothesized that since soiled bedding contains microbiota both from the feces and the cage environment, the bedding transfer SOP amongst multiple cages would reduce the variability of the gut microbiota among a large number of mice due to coprophagia. To test this hypothesis, the effect of this SOP on fecal microbiota variability was compared to the effects of the two conventional procedures using 16S rRNA gene amplicon sequencing and analysis. In addition, the impacts of possible confounding factors including time, diet, sex, and age on fecal microbiota using this SOP were assessed.

## MATERIALS AND METHODS

### Animal

This study protocol was approved by the University of Chicago Animal Care and Use Committee (protocols 71084 and 72101). C57Bl/6J mice were originally purchased from Jackson Laboratory (Bar Harbor, ME, USA). We prepared three cohorts for this study. For Cohort 1 and Cohort 2, mice were bred and raised at the University of Chicago Specific-Pathogen Free (SPF) Animal Vivarium. For Cohort 3, all mice were purchased from Jackson Laboratory at the same time and acclimated within our SPF animal vivarium for two weeks. Mice were fed Teklad Global 18% Protein Rodent Diet (2018) (Envigo, Madison, WI, USA) (Institutional Animal Care and Use Committee (IACUC) protocol 71084). Cohort 1 was switched to AIN-76A Purified Diet (*The American Institute of Nutrition, 1977*) (Envigo, WI, Madison, USA) (IACUC protocol 72101) between six and 12 weeks of age. Mice in Cohorts 2 and 3 were not switched to AIN-76A and used for

separate purposes after analyzing the base line gut microbiota at the onset of each study. To reduce batch effect of the diet lot, we purchased AIN-76A as one single which was used throughout the experiment. Teklad Global 18% Protein Rodent Diet (2018) was purchased by the University of Chicago Animal Resource Center, which distributes this diet to all vivarium rooms throughout the animal facility. Therefore, the variations between batches of Teklad Global 18% Protein Rodent Diet (2018) cannot be strictly excluded.

## Bedding transfer

Bedding transfers were performed among 14 female cages and 20 male cages (1–5 mice/cage) in Cohort 1. In our animal vivarium, fresh bedding is provided every 14 days by the animal husbandry staff. Bedding was mixed at 3–4 days and at 8–10 days following these cage changes, i.e., bedding transfers were performed twice within the two-week cycle. At each of these time points, roughly one-quarter of soiled bedding was collected from each cage and the bedding from all cages was mixed in an autoclaved sterile container, followed by redistribution across all cages. Soiled bedding was collected and mixed within a freshly cleaned biological safety cabinet (BSC) within the room where animals were housed. Standard barrier practices for BSC contamination include spraying down the hood with Clidox® and allowing it to sit for ∼3 min prior to turning it on, followed by wipe-down with clean paper towels. In addition to appropriate personal protective equipment (disposable gown, hair bonnet, and face mask), researchers also donned clean latex gloves and plastic tyvex sleeves. Only three personnel were involved in this SOP. Bedding transfer began at weaning and continued until the onset of fecal sample collections.

## Fecal samples

Fresh stool pellets were harvested from animals at designated time points. For chronological analysis in Cohort 1, all fecal samples were harvested at 6 am (time of lights on) throughout the study to avoid confounders of gut microbiota diurnal variation (*Leone et al., 2015*). Fecal samples were harvested one week after the cage changes throughout the experimental protocol and samples were kept frozen at −80 °C until DNA extraction.

## DNA extraction and 16S rRNA gene sequencing analysis

DNA was extracted from fecal samples using standard, published protocols (*Wang et al., 2009*). Sequences were obtained by MiSeq at the Next Generation Sequencing Core in the Biosciences Division at Argonne National Laboratory, amplifying the V4 region with standard protocols (*Earth Microbiome Project, 2016*). DNA sequences were analyzed by Quantitative Insights into Microbial Ecology (QIIME) version 1.9.1 (*Caporaso et al., 2010*), joining forward and reverse reads. Samples with less than 3,000 sequences were excluded from the analyses. Operational taxonomic units (OTUs) were picked at 97% sequence identity using *the GreenGenes Database (2013)*.

## Statistical analysis

Analysis of similarities (ANOSIM) was performed using QIIME to examine the difference between cohorts and the impact of sex on fecal microbiota. Given that the same mice were analyzed at multiple time points, and to control for these effects, ADONIS with strata

argument was performed using R, "vegan" package to assess the influence of diet and age. The number of permutations was 10,000 or the maximum number of permutations allowed by the data. Permutation test with 10,000 permutations was performed using R to compare the UniFrac distances of animals between cohorts. $p < 0.05$ was considered statistically significant.

### Accession numbers

The accession numbers for the mouse sample information and the microbial dataset reported in this paper are BioProject PRJNA397441 and SRA accession number SRP115420, respectively.

## RESULTS

### Bedding transfer reduces the variability of fecal microbiota between groups of mice

To examine if the bedding transfer protocol could reduce fecal microbiota variability in 16S rRNA amplicon sequencing analysis, three cohorts of C57Bl/6 wild-type mice were used (outlined in Fig. 1). Each cohort was maintained in a separate room in our animal vivarium under SPF conditions. For Cohort 1, five breeding pairs were obtained from an in-house bred mouse colony and maintained in one vivarium room ($F_0$ generation). Male and female mice were siblings, and their progeny ($F_1$ generation) were used to set up 25 breeding pairs. Mice from the $F_2$ generation (57 females and 57 males) were exposed to bedding transfer (Fig. 1A). For Cohort 2, 19 male mice from two separate in-house breeding colonies (12 and 7, respectively) were maintained in a separate SPF room and were analyzed as a single group (Fig. 1B). For Cohort 3, 36 male mice were purchased from Jackson Laboratory (Bar Harbor, ME, USA) and acclimated in individual cages in a third SPF room for two weeks (Fig. 1C). Animals in all cohorts were fed Teklad Global 18% Protein Rodent Diet (2018) (Envigo, Madison WI, USA) and were 6–12 weeks of age at the time of analyses. Cohort 1 males were compared to those in Cohorts 2 and 3, respectively (Fig. 2). PCoA plots for Cohorts 1 and 2 are presented in Fig. 2A, where unweighted UniFrac distances describe the OTUs existing in samples, while weighted UniFrac distances take into account the proportions of those OTUs. Permutation test for both unweighted and weighted UniFrac distances showed that the within-group distances of mice in Cohort 1 were significantly decreased as compared to mice in Cohort 2 ($p = 0.0001$ and 0.0001 in unweighted and weighted UniFrac distances, respectively; Fig. 2B). ANOSIM showed significant differences in fecal microbiota between Cohorts 1 and 2 ($p = 0.001$ and $R = 0.778$ in unweighted UniFrac distances, $p = 0.001$ and $R = 0.655$ in weighted UniFrac distances). A comparison of unweighted and weighted UniFrac distances of Cohorts 1 and 3 also revealed significant decreases of within-group Unifrac distances (each $p = 0.0001$; Fig. 2D) in Cohort 1 as compared to Cohort 3. ANOSIM showed significant differences in fecal microbiota between Cohorts 1 and 3 ($p = 0.001$ and $R = 0.989$ in unweighted UniFrac distances, $p = 0.001$ and $R = 0.960$ in weighted UniFrac distances; Fig. 2C). The bacterial community memberships based on 16S rRNA sequencing at both the phylum and genus levels in each cohort are shown in Fig. S1. The comparisons of microbial community

**A Cohort 1**

**F0 generation**  ♀ & ♂  ♀ & ♂  ♀ & ♂  ♀ & ♂  ♀ & ♂
**5 breeding pairs**

**F1 generation**  ♀ & ♂  ♀ & ♂  ♀ & ♂  ····· ♀ & ♂  ♀ & ♂
**25 breeding pairs**

**F2 generation**  57 female and 57 male mice

Bedding transfers from weaning until analysis

**Analysis**

**B Cohort 2**

| Breeding colony 1 | Breeding colony 2 |

| 12 male mice | 7 male mice |

**19 subjects in total**

**Analysis**

**C Cohort 3**
Purchased
36 male mice

Acclimation for 2 weeks

**Analysis**

**Figure 1** **Murine cohort design.** All mice in Cohorts 1–3 were on a C57Bl/6J background, maintained on corncob bedding under standard housing conditions and fed Teklad Global 18% Protein Rodent Diet unless indicated otherwise. (A) Cohort 1—Five breeding pairs were prepared from mice housed in one room ($F_0$ generation). Twenty-five breeding pairs were set up from the $F_0$ progeny ($F_1$ generation). Fifty-seven female and 57 male mice of $F_2$ generation were used for analysis of Cohort 1. Bedding was mixed among all $F_2$ cages in Cohort 1 beginning at weaning (three weeks old) until the diet switch to AIN-76A (6–12 weeks old). All mice were housed in one room. (B) Cohort 2—Twelve and seven age-matched male mice (19 total, 10–12 weeks old) were obtained from two separate C57Bl/6 breeding colonies, respectively which were housed in a single, yet separate room than Cohort 1. (C) Cohort 3—Thirty-six age-matched male mice (8–10 weeks old) were purchased from the Jackson Laboratories and acclimated for two weeks in a single, yet separate room than both Cohorts 1 and 2.

membership between cohorts housed in separate rooms within a vivarium by itself can lead to differences in gut microbiota that could be potential confounders. Daily health check monitoring by the animal husbandry staff and our own weekly health assessment of each mouse revealed no apparent and observable adverse events, such as ruffled fur appearance, decreased activity, injury due to fighting, or alopecia. These animal welfare concerns are

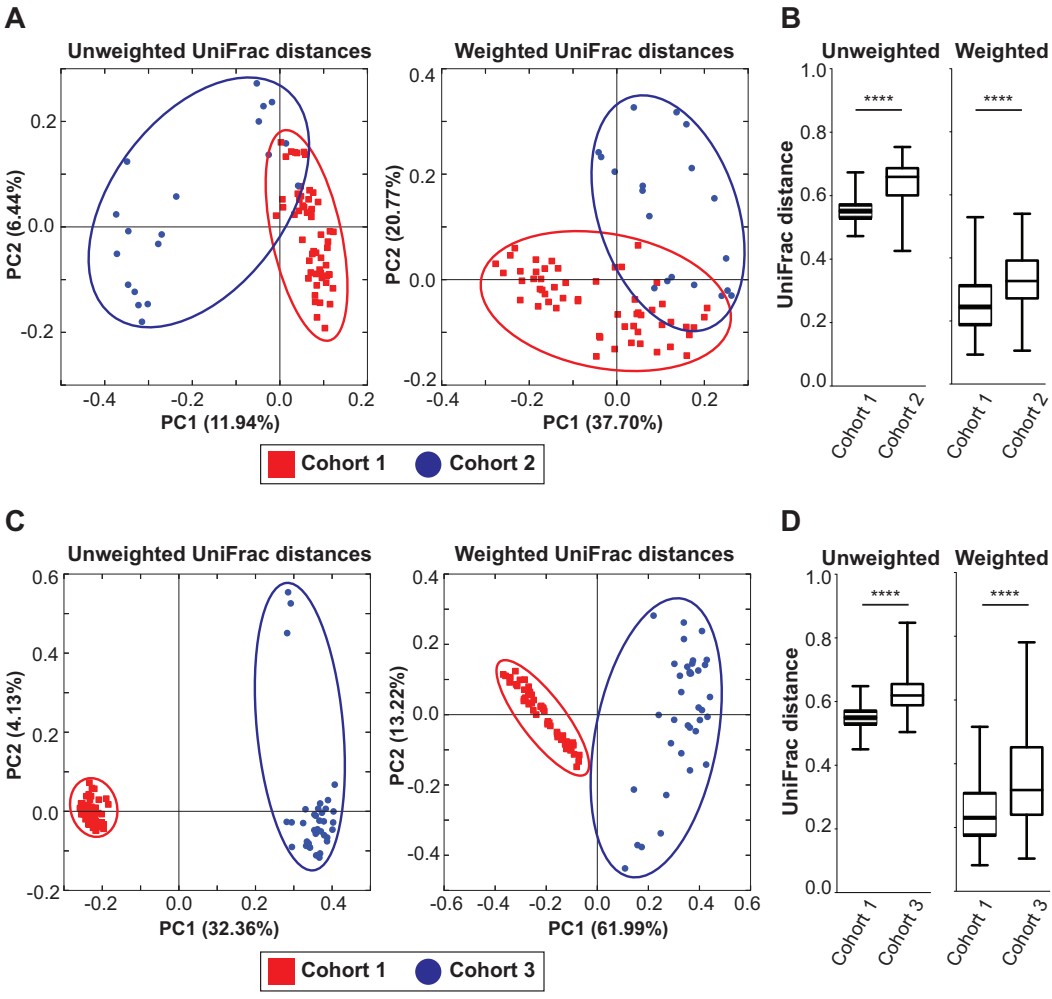

**Figure 2  Mixed bedding transfer reduces the variability of fecal microbiota among mice.** (A) PCoA plots of both unweighted and weighted UniFrac distances of 16S rRNA amplicon sequences from fecal samples obtained from Cohort 1 males (red squares) and those of Cohort 2 males (blue circles). (B) The unweighted and weighted UniFrac distances between samples within Cohort 1 and those within Cohort 2 were compared. (C) PCoA plots of both unweighted and weighted UniFrac distances for samples of Cohort 1 (red squares) and those of Cohort 3 (blue circles). (D) The unweighted and weighted UniFrac distances between samples within Cohort 1 and those within Cohort 3 were compared. **** $p = 0.0001$.

in particular a limitation for co-housing male mice from different litters, which is avoided using this SOP. Together, these findings show that our SOP minimizes variance among animals without adverse co-housing effects when a large study cohort obtained from several breeding pairs is required. To examine how this SOP impacted downstream experimental procedures, we focused all subsequent analyses on Cohort 1.

## Normalization of starting gut microbiota results in a reproducible impact of diet on murine fecal microbiota community membership

The 57 female and 57 male mice in Cohort 1 were weaned and maintained on Teklad Global 18% Protein Rodent Diet throughout the bedding transfer SOP and baseline fecal

samples were collected. Bedding transfer was stopped, and mice were switched to a defined AIN-76A Purified Diet formulation (*The American Institute of Nutrition, 1977*) (Envigo, Madison, WI, USA). One week after starting AIN-76A, fecal samples were harvested and DNA was extracted and compared with the baseline samples. We compared 16S rRNA gene amplicon sequencing separately for females and males. Fifty-six female and 56 male samples from the first harvest and 51 female and 57 male samples from the second harvest were plotted in PCoA plots (Fig. 3, samples with less than 3,000 reads were dropped from the analysis). PCoA analysis revealed both female and male samples exhibited significant shifts of microbiota in both unweighted and weighted UniFrac distances with ADONIS ($p = 0.0001$ and $R^2 = 0.157$ in female unweighted UniFrac distances, $p = 0.0001$ and $R^2 = 0.405$ in female weighted UniFrac distances, $p = 0.0001$ and $R^2 = 0.158$ in male unweighted UniFrac distances, $p = 0.0001$ and $R^2 = 0.399$ in male weighted UniFrac distances). This data reveals that diet switch from Global 18% Protein Diet to AIN-76A altered the fecal microbiota dramatically in both sexes in one week. The bacterial community membership based on 16S rRNA gene amplicon sequencing at both the phylum and genus levels of female and male animals fed each diet are shown in Fig. S2.

### Sex differences in fecal microbiota are evident despite bedding transfer amongst sexes

Eight female and eight male mice in Cohort 1 were continued on the AIN-76A diet for an additional 24 weeks after the initial analysis to examine the influence of switching diet described above. At Week 0 (the initial analysis), the mice were 7–10 weeks old and fecal samples were analyzed at week 12 and week 24. PCoA plots of unweighted and weighted UniFrac distances at Weeks 0, 12 and 24 are shown in Figs. 4A–4C. At all time-points, ANOSIM showed a significant difference between female and male fecal 16S-rRNA-based community structure (*Week 0*: $p = 0.001$ and $R = 0.432$ in unweighted UniFrac distances and $p = 0.001$ and $R = 0.397$ in weighted UniFrac distances; *Week 12*: $p = 0.001$ and $R = 0.465$ in unweighted UniFrac distances and $p = 0.004$ and $R = 0.277$ in weighted UniFrac distances; *Week 24*: $p = 0.003$ and $R = 0.695$ in unweighted UniFrac distances and $p = 0.008$ and $R = 0.319$ in weighted UniFrac distances). These results indicate that there was a persistent sex difference in fecal microbiota despite the bedding transfer protocol. However, we did observe several common core OTUs in both sexes. The bacterial community membership at both the phylum and genus levels of female and male animals at Weeks 0, 12 and 24 are shown in Fig. S3.

### Aging elicits a large impact on fecal microbiota community membership regardless of sex

The eight females and eight males were also analyzed to investigate the effect of aging on the fecal microbiota. PCoA plots of unweighted and weighted Unifrac distances with samples at Weeks 0, 12 and 24 are shown in Fig. 5. ADONIS demonstrated that there were shifts in gut microbial community membership associated with aging in both sexes ($p = 0.0001$ and $R^2 = 0.189$ in female unweighted UniFrac distances, $p = 0.0001$ and $R^2 = 0.362$ in female weighted UniFrac distances, $p = 0.001$ and $R^2 = 0.142$ in male unweighted UniFrac distances, $p = 0.0004$ and $R^2 = 0.396$ in male weighted UniFrac distances). ADONIS was

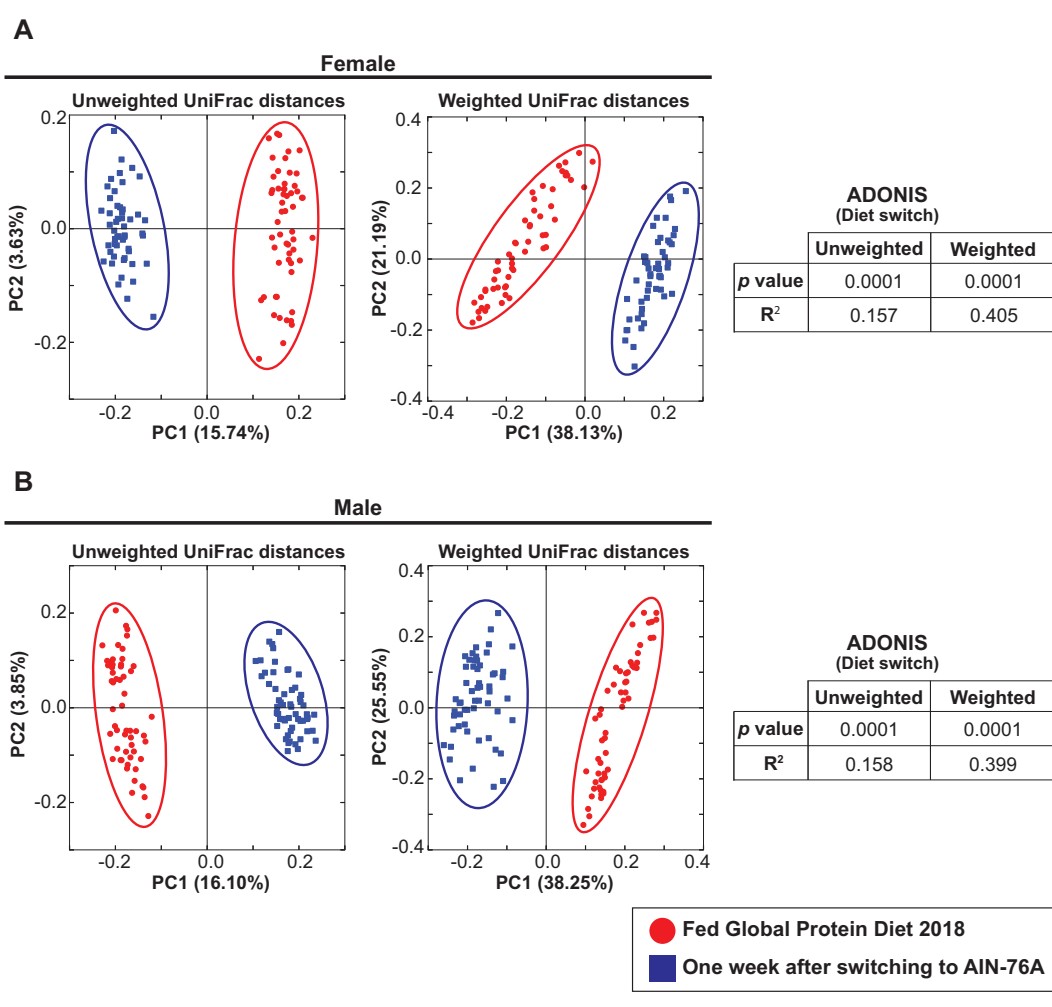

**Figure 3 Dietary switch results in a shift of the fecal microbiota composition.** Comparison of the fecal microbiota of 57 female mice (A) and 57 male mice (B) in Cohort 1 when fed Teklad Global 18% Protein Rodent Diet (red circles) and the same mice at 1 week after switching the diet to AIN-76A Purified Diet (blue squares). PCoA plots of unweighted and weighted UniFrac distances are shown.

also performed to assess $R^2$ values in comparisons between Weeks 0 vs. 12 and Weeks 12 vs. 24. The higher $R^2$ is interpreted as the greater amount of variance among the observations explained by the time point difference. The test suggested that the aging effect was most evident in the first 12 weeks in males (*Weeks 0 vs. 12*: $R^2 = 0.125$ in unweighted UniFrac distances and $R^2 = 0.418$ in weighted UniFrac distances; *Weeks 12 vs. 24*: $R^2 = 0.0676$ in unweighted UniFrac distances and $R^2 = 0.0749$ in weighted UniFrac distances) while the aging effect was evident in both the first and later 12 weeks in females (*Weeks 0 vs. 12*: $R^2 = 0.134$ in unweighted UniFrac distances and $R^2 = 0.202$ in weighted UniFrac distances; *Weeks 12 vs. 24*: $R^2 = 0.132$ in unweighted UniFrac distances and $R^2 = 0.303$ in weighted UniFrac distances).

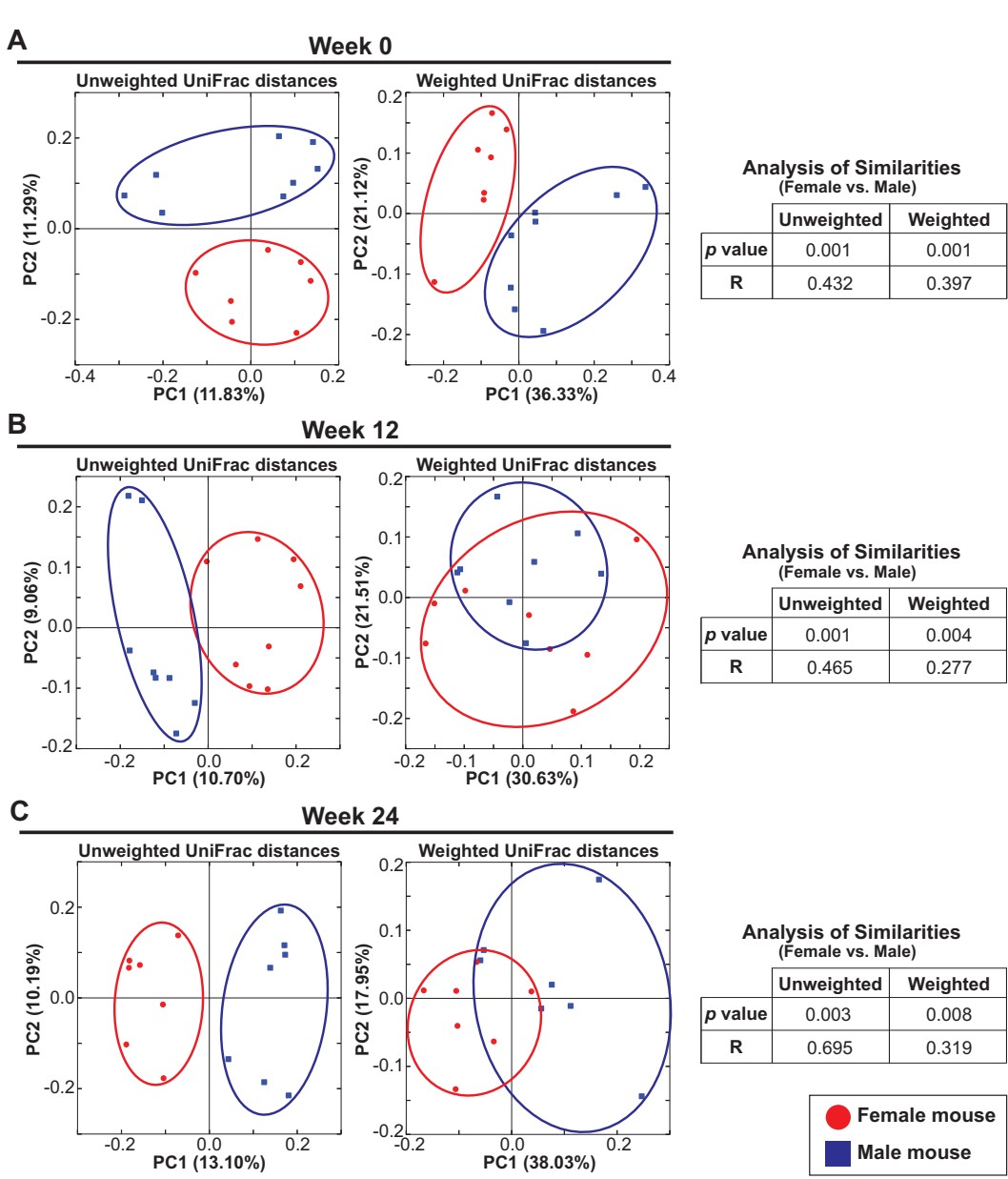

**Figure 4 Sex drives gut microbial assemblage independently of mixed bedding transfer that persists over time.** Fecal microbiota of eight female and eight male mice (7–10 weeks old) was tracked over time for 24 weeks. Unweighted and weighted UniFrac distances were analyzed at Weeks 0, 12 and 24 to compare female and male samples. PCoA plots present female (red circles) and male (blue squares) samples at Week 0 (A), Week 12 (B) and Week 24 (C).

## DISCUSSION

Murine models are commonly used to study host-microbe-environmental interactions and many investigators assume there is uniformity of gut microbiomes within groups and cages of mice prior to an experiment. Unfortunately, this is often not the case and even in studies conducted in a single room (environment), large variations in gut microbiota

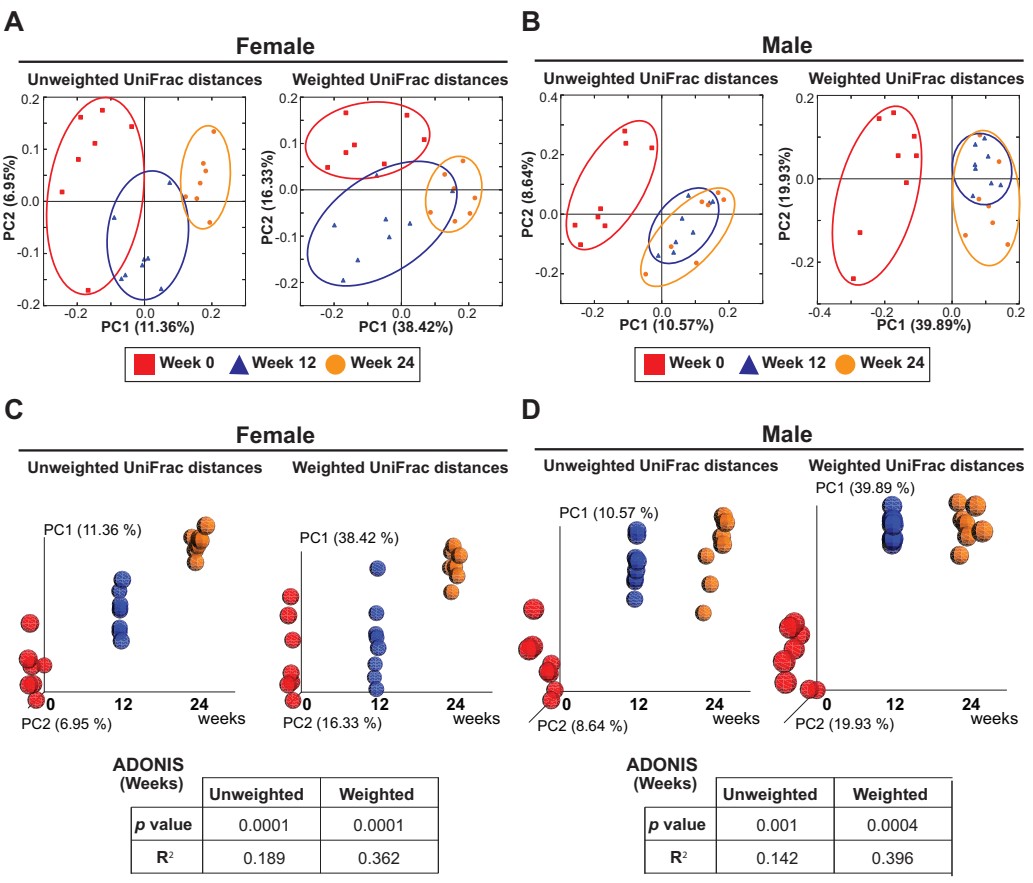

**Figure 5** **Aging is an independent driver of microbial assemblage, particularly in female mice.** (A and B) PCoA plots of unweighted and weighted UniFrac distances of fecal microbiota at Weeks 0 (red squares), 12 (blue triangles) and 24 (orange circles) in each gender. (C and D) PCoA plots with a time axis in each gender are shown.

exist between cages that can significantly affect reproducibility and skew experimental results. These differences can arise from multiple factors, including genetic background, diet, sex, and age (*Hoy et al., 2015*). Controlling these variables to insure that the starting microbiota is uniform among paired experimental groups is therefore essential to improve rigor and reproducibility, which are now mandated by federal and non-federal funding agencies (*National Institutes of Health, 2017*). Furthermore, this effort can contribute to the concept of the 3Rs (Replacement, Reduction, and Refinement), which is crucial from an ethical and scientific viewpoint and is now globally considered in animal studies. Other considerations prior to initiating studies of the gut microbiome include the development of an effective and practical study design, the necessity of starting with sufficiently large cohorts if time-series tissue harvests are envisioned, and the sufficient statistical power for reliable data analysis. To address this, we established a non-invasive bedding transfer SOP that promotes uniformity of gut microbiota across cages and this SOP also considers commonly used procedures to minimize confounding factors across cohorts as well as normalized microbiota prior to experiments. These involve: (1) using a single genetically

identical strain, (2) housing all animals in one specific room (to control environment), (3) using the same diet for all mouse groups, (4) controlling for the effects of aging by utilizing age-matched mice, and (5) separating sexes and performing these analyses separately. Despite these measures, significant differences in gut microbiota along with generational drift are observed. While littermate cohorts are desirable, many studies require a large number of mice at the start to account for attrition in study cohorts as animals are harvested in sufficient numbers to adequately power the data at each time point. Here, investigators have to either combine different breeding colonies or acclimate age-matched mice purchased from a single vendor to achieve these numbers before the initiation of experiments. The logistics of achieving these pre-experimental conditions can be challenging and inherently result in differences in microbial assemblage among unique study groups and cages. We now report a practical and effective bedding transfer protocol that addresses these problems, reducing many of the variabilities in gut microbiota among animals in large cohort studies of the gut microbiome. However, we concede that the present study design has a limitation to assess the efficacy of bedding transfer SOP because a strict control group without the SOP was not included. We also note that host factors, such as sex and aging are independent determinants of microbial assemblage, which must be factored into all studies of this nature. With regard to the former, this SOP can be applied safely even to co-housed adult male mice across multiple cages, completely avoiding territorial problems such as fighting and injury. Importantly, clear shifts in gut microbial communities in response to the transition from mouse chow to a purified diet is an excellent example of how a strong effect size can be seen across the different groups.

Differences in bacterial community membership were observed between Cohorts 1–3 housed in separate rooms. We speculated two potential reasons for the observed "room effect". First, subtle differences in room environments (temperature, staff handlers, ventilation, etc.) can impact the assemblage of gut microbial communities among mice housed. Another possibility is that genetic drift among the C57Bl/6J mice used for breeding in different rooms (either in-house or within a vendor's facility) could influence microbiota community membership. Regardless of the observed differences in gut microbiota of mice housed in different rooms, our conclusions are not altered and show that our SOP is effective and practical. With regard to 16S rRNA amplicon data analysis, we recognize the OTU-based methodology that Quantitative Insights into Microbial Ecology (QIIME) (*Caporaso et al., 2010*) employs has a limitation, including the reference databases (*Schloss & Westcott, 2011*). The limitation of databases is most apparent in the assignment of taxonomy, which does not provide sufficient resolution beyond the genus level and therefore cannot rule out subtle changes in gut microbiota at the species or strain level. Furthermore, the 16S rRNA amplicon analysis only provides information on community membership and is unable to provide insights into community function and the impact elicited by diet, sex, and aging. Further analysis using shotgun metagenomics, metatranscriptomics, metabolomics and metaproteomics would be needed to address these issues which is beyond the scope of this study.

In regard to the impact of diet, both female and male mice showed a dramatic change in microbiota community membership after only one week following diet switch, which

has also been observed by others in mice (*Liu et al., 2012*; *Turnbaugh et al., 2009b*). This phenomenon of rapid changes of gut microbiota induced by diet has also been reported in humans (*David et al., 2014*). Together, these findings underscore that diet is an important determinant of gut microbial assemblage even for short-term studies. In this regard, each dietary component has to be carefully considered, as well as the source of each nutrient used for diet preparation. It cannot be assumed, for instance, that all dietary fats are the same. Teklad Global 18% Protein Rodent Diet contains soybean oil, whereas AIN-76A Purified Diet contains corn oil in different percentages; similarly, the fiber component is also dramatically different between these two diets. Even more subtle differences in diet (i.e., micronutrient content) can dramatically impact the gut microbiota.

We noted strong sex differences among the groups even after the mixed bedding protocol was instituted, suggesting that sex is a strong independent host factor driving gut microbial assemblage. This finding emphasizes the importance of analyzing both female and male animals separately in all gut microbiome studies, as these differences could cause differential effects in the host, e.g., the development of the immune system (*Brown, Sadarangani & Finlay, 2013*; *Gensollen et al., 2016*) or in host xenobiotic metabolism (*Claus et al., 2011*; *Meinl et al., 2009*; *Nobutani et al., 2017*). In light of well-established differences in treatment regimens and drug toxicity among male and female human subjects (*Soldin & Mattison, 2009*), inclusion of both sexes in studies examining gut microbes is now encouraged by the National Institutes of Health and other funding agencies.

Finally, we observed a microbiome drift associated with aging, underscoring the importance of starting with murine cohorts at a similar age, as well as using age-matched control groups followed in parallel with test groups. Interestingly, female mice showed progressive changes in gut microbiota even after Week 12 that was observed through Week 24 of the study, while similar changes in males during this period were less apparent. Considering the ages of these mice at the time of analyses (19–22 weeks of age at week 12 and 31–34 weeks of age at week 24), we speculate that age-dependent shifts in female hormones could have contributed to the observed drift in gut microbiota (*Org et al., 2016*). Indeed, the pregnancy rate in C57Bl/6J female mice after 24 weeks of age is generally very low, possibly due to age-related hormonal changes that affect fertility. Diurnal variation in gut microbiota (*Leone et al., 2015*; *Thaiss et al., 2014*; *Zarrinpar et al., 2014*) underscores the importance of harvesting fecal samples at the same time in a day when repeat collections are required. Furthermore, a recent report suggested that the timing of the last cage change before sampling also may be a potential source for bias in murine microbiome studies (*Rodriguez-Palacios et al., 2018*). These factors should be taken in to consideration when a study is designed. To overcome this, we collected fecal samples at 6 am at one week after the cage change throughout the entire study period.

Further studies are necessary to refine and identify limitations to our outlined SOP. For instance, given the soiled bedding is exposed to oxygen there is a possibility that this SOP is skewed towards the successful transfer of aerobic bacteria rather than strict anaerobes. However, a similar bias may also occur in the co-housing procedure. Furthermore, with gavage procedure, the passing of microbes through the gastrointestinal (GI) tract itself

could result in another selection bias in the lower GI tract. We believe that it is an important aspect to assess the limitations across all of these procedures in future studies.

## CONCLUSIONS

Our bedding transfer SOP is practical and effective in reducing variability of fecal microbiota amongst individual mice when a large study cohort study is required. We also report that aging, sex, and time of fecal sampling are independent variables of microbial assemblage that should be taken into consideration when undertaking studies of this nature in mice. This SOP along with consideration of additional host drivers of the gut microbiota community membership and function may improve the quality of future murine studies of the gut microbiome.

## ACKNOWLEDGEMENTS

We thank Mrinalini Rao, PhD for help in editing the manuscript. We also thank Masahiro Yamamoto, PhD for his support in the experiments. We appreciate the consulting program provided by Department of Statistics, The University of Chicago and thank Tae Kim and her team for their great help in statistical analyses.

### Funding

The present research was supported by the NIH Digestive Disease Core Research Center (NIDDK P30DK42086). Additional support was provided through Tsumura & Co. The funders had no role in study design, data collection and analysis, decision to publish, or preparation of the manuscript.

### Grant Disclosures

The following grant information was disclosed by the authors:
NIH Digestive Disease Core Research Center: NIDDK P30DK42086.
Tsumura & Co.

### Competing Interests

A. Murat Eren is an Academic Editor for PeerJ.

### Author Contributions

- Jun Miyoshi conceived and designed the experiments, performed the experiments, analyzed the data, contributed reagents/materials/analysis tools, prepared figures and/or tables, authored or reviewed drafts of the paper, approved the final draft.
- Vanessa Leone conceived and designed the experiments, performed the experiments, analyzed the data, contributed reagents/materials/analysis tools, authored or reviewed drafts of the paper, approved the final draft.
- Kentaro Nobutani and Kristina Martinez-Guryn conceived and designed the experiments, performed the experiments, contributed reagents/materials/analysis tools, approved the final draft.

- Mark W. Musch conceived and designed the experiments, performed the experiments, authored or reviewed drafts of the paper, approved the final draft.
- Yunwei Wang and Alexandria M. Bobe performed the experiments, approved the final draft.
- Sawako Miyoshi performed the experiments, contributed reagents/materials/analysis tools, approved the final draft.
- A. Murat Eren and Eugene B. Chang conceived and designed the experiments, authored or reviewed drafts of the paper, approved the final draft.

## Animal Ethics

The following information was supplied relating to ethical approvals (i.e., approving body and any reference numbers):

This study protocol was approved by the University of Chicago Animal Care and Use Committee (protocols 71084 and 72101).

## Data Availability

The accession numbers for the mouse sample information and the microbial dataset reported in this paper are BioProject PRJNA397441 and SRA accession number SRP115420, respectively.

## Supplemental Information

Supplemental information for this article can be found online at http://dx.doi.org/10.7717/peerj.5166#supplemental-information.

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
