# Peer review of "Minimizing confounders and increasing data quality in murine models for studies of the gut microbiome"

_PeerJ, doi:10.7717/peerj.5166_

## Round 0.1 · original submission · Major Revisions

Both reviewers found the work important. However, they have some concerns about the validity of the study design and hence the conclusions drawn from it. I also echo Reviewer 2's comment about the cage-based analysis. It is important to show that the cage-to-cage variation has been largely diminished by using the proposed SOP to support their major conclusion. Please also pay attention to the statistical tests used to compare distances between two groups (e.g. Figure 2B,D). The within-group distances are correlated and Mann-Whitney U test is not valid. Consider using permutation test.

Reviewer 1 ·

Basic reporting

This is a nice article on a small, yet important, topic for the improvement of mouse studies with consideration to the microbiome as an experimental variable. This is an important contribution to the field where there is a lack of standards. Dirty bedding transfer is definitely an approach many use to minimize microbiome variability so it is highly relevant with data demonstrating the ability of the method to decrease variation. Going forward the field needs more rigorous investigation of the potential adverse effects of the method, as this is not thoroughly investigated here. Nice and clear figures.
Generally, very well-written, clear and concise language, easy to follow and understand and this is also supported by the figures.
I have three major concerns and some minor comments.

Major:
1.
Line 187-189:
How did you monitor adverse affects on animal behavior? Was it based subjective on feedback from the technicians or did you monitor in a controlled way and compare to previous data on injuries etc.?
The study lacks a more controlled way of monitoring the potential adverse effects and I would advise against concluding that there are no adverse effects of the method without proper data to back it up, e.g. corticosterone measurements (e.g. in feces), behavioral testing, infrared thermography or even just BW monitoring. I think these consideration and caveats should be described in the manuscript and reflected in the conclusions.
2.
Title + line 346 + figure 3: how can it be claimed that this method can improve reproducibility without testing this? You have observed the decreased variation. You show that the method with decreasing the variation can indeed lead to a big effect size, e.g. of a dietary intervention. But from what I understand you have not made repeated experiments, showing that this a reproducible method?
Title: Also, how was it concluded that data quality is maximized? The effect size seen in the dietary intervention was not compared against a control group without dirty bedding transfer?
3. See comment under Experimental Design

Minor:
Line 63-64: Hoy et al. did not define the “cage effect” or were the first ones to describe it. Thus, it is more correct to write something like “…., variation between cages can be observed, a phenomenon called the “cage effect”” and cite: Hildebrand et al, Genome Biol 2013., McCafferty et al 2013, Hoy et al 2015.
Line 87-88: “…, appreciable differences in starting gut microbiota have been observed.” Can you provide a reference for that?
Line 97: I would use “namely” instead of “viz.” to make it easier to follow for a broad audience.
Line 125: How was bedding sampled? Aseptically?
Line 130: Write here time of day of the fecal samplings. Did you consider timing of sampling in relation to when the last cage change had happened? This can potentially introduce bias as recently published: Rodriguez-Palacios A, Aladyshkina N, Ezeji JC, Erkkila HL, Conger M, Ward J, et al. “Cyclical Bias” in Microbiome Research Revealed by A Portable Germ-Free Housing System Using Nested Isolation. Sci Rep. 2018;8:3801. doi:10.1038/s41598-018-20742-1.
Line 216: I was once told by a native English reviewer that the term “gender” is for humans and “sex” should be used for animals. Just a suggestion, I know many papers use the term gender when referring to mice.
Line 216: There is a typo: transfer -> transfer
Line 216-230: It is interesting to see that there are strong sex differences despite exchanging of bedding between the sexes. That really underlines that the sex effect on the microbiome is quite strong.
Line 263-264: The sentence is US-centric. Consider a more generic verbiage. As an example, consideration of the 3Rs (to which reproducibility is inherent) has been mandated by EU regulations for years.
Line 272: Did you use the same batch of diet? For chow diets, there may be seasonal batch variation potentially affecting microbiome
Line 286: Is there a typo: co-house -> co-housed?
Line 285-287: I don’t fully understand the sentence. The male mice were single-housed, from what I understood?
Line 290: I would usually not refer to a figure in the discussion
Line 290-297: Isn’t is obvious that the microbiomes of the three cohorts would differ? They are not originating from the exact same source. I know they originally all came from Jax, but did they come from the same barrier there? And then there is of course the facility and room specific environmental impact. So not surprising that there is a room effect.
Line 298: Isn’t it more a limitation of the databases used than the software QIIME per se?
Line 306-316: Please comment on why you did not do the diet intervention in cohort 2 or 3. That would have been relevant to see if the effect size would actually be bigger in cohort 1.

Experimental design

The study design is clear to understand (both text and the figure).
Yet, the setup is somewhat flawed by the fact that cohort 1 and 2 differ in more aspects than the dirty bedding transfer protocol. Ideally, these two cohorts should have been obtained/bred exactly the same way and dirty bedding transfer only performed in one of the groups. This must be commented on and addressed in the manuscript.

Validity of the findings

No concerns, besides the comments I have already made. The statistics and methods used are appropriate.

Additional comments

Are there organisms known not to transfer via the dirty bedding? If some, potentially important species do not transfer this way the method may be less reliable for studies where these organisms are imperative.

·

Basic reporting

No comment.

Experimental design

The research addresses an important issue in the field: cage-to-cage variation fecal microbiota observed in mouse studies. The comparisons between cohorts (results section 1) are especially relevant and will be useful in designing future studies.

In section 2 of the results, the impact of diet is assessed in cohort 1 before and after diet switch. However, there was no control group of mice not receiving the new diet (AIN-76A), and one could argue that the observed changes were due to drift over time alone. Are you able to point to earlier published results or make another argument in response to this possibility?

Validity of the findings

The purpose of the SOP presented in this paper is to reduce cage-to-cage variation prior to the start of the experiment. However, you do not directly analyze the cage-to-cage variation. Instead, the within-group distances are used as a surrogate measure: the dispersion within a cohort decreases due to increased cage-to-cage differences.

The conclusions would be much better supported by a statistical analysis of the cage-to-cage differences. Based on UniFrac distances, I would expect to see no difference in within-cage similarity, but see a decrease in between-cage distances when comparing cohort 1 to cohort 3.

This applies to all sections of the results. In section 2, did the dietary intervention alter between-cage/within-cage distances? In section 3, how large were sex-based differences relative to differences between and within cages? In section 4, did cage-to-cage differences increase over time as the mice aged?

The cages should also be marked on the figures for inspection (maybe in supplemental). For example, in Figure 1C, do the samples in the upper right come from the same cage?

As a secondary issue, the authors should be careful to treat repeated measures properly in the statistical tests. If the same mouse is measured twice, the values should not be considered as independent samples. In the ANOSIM test, I believe that the "strata" keyword argument can be used to handle repeated measures (though I'm curious to see if people disagree with me on that).

Additional comments

Minor comments:

In section 3 of the results, did the sex-based differences match previous research?

In section 4 of the results, it seems to me that the analysis of PCoA axis 1 and 2 doesn't buy you much. What is the advantage of showing a correlation with a specific axis? Although repetitive, I would recommend sticking to the distance-based tests unless the correlation with axis 1/axis 2 has special importance.

Part of the motivation for researchers to adopt the SOP would be to increase power in mouse studies (by decreasing the within-group variance). To really drive this point home, I think you are in a position to estimate how much the SOP increased power in the experiments presented here. That number would be very relevant to researchers planning future studies.

In section 1 of the results, it would increase clarity to say "significantly decreased within-group distances" rather than "significantly decreased distances." I also discourage the "clustering" language unless you actually show clustering results.

---

## Round 0.2 · accepted · Accept

The authors have addressed the comments adequately and toned down the findings by changing the title. The SOP will be very useful to the community.

#